Genetic divergence between populations of feral and domestic forms of a mosquito disease vector assessed by transcriptomics

Price Dana C. d.price@rutgers.edu
Fonseca Dina M. dinafons@rci.rutgers.edu
Department of Entomology, Rutgers University , New Brunswick, NJ , USA
Schwander Tanja
Electronic publication date: 2015 Feb 26
Publication date: 2015
Volume: 3
Electronic Location ID: e807
Received 2014 Nov 13; Accepted 2015 Feb 9
Copyright: © 2015 Price and Fonseca
Copyright year: 2015
Copyright holder: Price and Fonseca
License: This is an open access article distributed under the terms of the Creative Commons Attribution License, which permits unrestricted use, distribution, reproduction and adaptation in any medium and for any purpose provided that it is properly attributed. For attribution, the original author(s), title, publication source (PeerJ) and either DOI or URL of the article must be cited.
License URL: https://creativecommons.org/licenses/by/4.0/

Keywords: Culex pipiens complex, Molestus, Cx. quinquefasciatus, Mosquito, Natural selection, Ka/Ks, Cryptic species, Domestication, Arthropod vector

Funding: New Jersey Mosquito Control Association NE-1043 Multistate This work was funded by a New Jersey Mosquito Control Association Daniel M. Jobbins scholarship to Dana C. Price and by start-up and NE-1043 Multistate funds to Dina M. Fonseca. The funders had no role in study design, data collection and analysis, decision to publish, or preparation of the manuscript.

==============================
Culex pipiens, an invasive mosquito and vector of West Nile virus in the US, has two morphologically indistinguishable forms that differ dramatically in behavior and physiology. Cx. pipiens form pipiens is primarily a bird-feeding temperate mosquito, while the sub-tropical Cx. pipiens form molestus thrives in sewers and feeds on mammals. Because the feral form can diapause during the cold winters but the domestic form cannot, the two Cx. pipiens forms are allopatric in northern Europe and, although viable, hybrids are rare. Cx. pipiens form molestus has spread across all inhabited continents and hybrids of the two forms are common in the US. Here we elucidate the genes and gene families with the greatest divergence rates between these phenotypically diverged mosquito populations, and discuss them in light of their potential biological and ecological effects. After generating and assembling novel transcriptome data for each population, we performed pairwise tests for nonsynonymous divergence (Ka) of homologous coding sequences and examined gene ontology terms that were statistically over-represented in those sequences with the greatest divergence rates. We identified genes involved in digestion (serine endopeptidases), innate immunity (fibrinogens and α-macroglobulins), hemostasis (D7 salivary proteins), olfaction (odorant binding proteins) and chitin binding (peritrophic matrix proteins). By examining molecular divergence between closely related yet phenotypically divergent forms of the same species, our results provide insights into the identity of rapidly-evolving genes between incipient species. Additionally, we found that families of signal transducers, ATP synthases and transcription regulators remained identical at the amino acid level, thus constituting conserved components of the Cx. pipiens proteome. We provide a reference with which to gauge the divergence reported in this analysis by performing a comparison of transcriptome sequences from conspecific (yet allopatric) populations of another member of the Cx. pipiens complex, Cx. quinquefasciatus.

Introduction

Specific life-history traits of arthropod disease vectors can determine the duration and severity of outbreaks by influencing vectorial capacity (NAS 2008). Plasmodium falciparum, the deadliest of human malaria agents, Wuchereria bancrofti, the widespread causative agent of lymphatic filariasis, and both dengue and yellow fever viruses are transmitted by mosquito vectors that live in close association with and feed near-exclusively on humans. Anthropophilic mosquito phenotypes maximize transmission rates and promote high pathogen virulence of these diseases (Dieckmann et al., 2002). In contrast, zoonotic diseases requiring amplification cycles in non-human vertebrate hosts such as West Nile virus or eastern equine encephalitis will only spill over to humans (often to the detriment of the parasite and the human) if a vector with a broader range of hosts becomes involved (Farajollahi et al., 2011; Kilpatrick et al., 2006). Although blood meal analyses have demonstrated strong associations between vector species and suites of vertebrate hosts, the mechanisms underlying host-choice are still broadly unknown and are often ascribed to environmental instead of genetic causes (Chaves et al., 2010).

The northern house mosquito, Culex pipiens, is comprised of two morphologically indistinguishable forms (eco/biotypes), Cx. pipiens form pipiens L. and Cx. pipiens form molestus Forskål (herein f. pipiens and f. molestus, respectively). Despite their morphological identity and very close phylogenetic history (Fonseca et al., 2004b), the two forms exhibit notable ecological and behavioral differences that make their identification possible. The feral form, f. pipiens requires a vertebrate bloodmeal for all egg development (anautogeny), enters winter diapause when ambient light levels decrease below a locally pre-established threshold in the fall (heterodynamous), swarms as a prelude to mating (eurygamous), and is primarily ornithophilic. In contrast, f. molestus can forego a bloodmeal for its first gonotrophic cycle (autogeny), adults remain gonoactive during winter months (homodynamous), which means they are often restricted to subterranean environments with standing water such as subways and sewers (hypogeous) that remain warm. Males of f. molestus will mate in very confined spaces (stenogamous) and females frequently feed on mammals, including humans (references summarized in Fonseca et al. (2004a)); see Gomes et al. (2012) for latest blood meal studies). Cx. pipiens f. molestus is a worldwide invasive species, spread by humans to all continents except Antarctica (Farajollahi et al., 2011) while f. pipiens has remained restricted to Northern Europe. Cx. pipiens populations within the United States are hybrids of the two forms (Fonseca et al., 2004b; Strickman & Fonseca, 2012) and are implicated in the maintenance and transmission of epizootic arboviruses such as West Nile Virus (WNV) to humans resulting in illness and occasionally death (Kramer, Styer & Ebel, 2008).

The two forms of Cx. pipiens are very closely related, as is evident from their identical morphology and genetic similarity (Fonseca et al., 2004b). This has led to controversy over their taxonomic standing (Harbach, Harrison & Gad, 1984; Spielman et al., 2004). However, they are differentiated at hyper-variable loci such as the flanks of microsatellites (Bahnck & Fonseca, 2006) indicating recent separate evolutionary histories. The genetic similarity despite striking differences in ecology, behavior and physiology indicate that f. molestus may have diverged from f. pipiens and evolved its association with humans as recently as 10,000 years ago (Fonseca et al., 2004b). This recent split represents an exceptional opportunity to test whether targets of molecular evolution in Cx. pipiens mosquitoes can be elucidated using two phenotypically diverged populations. Additionally, by framing the results in context of phenotype, the data generated would serve as a first look at the molecular basis for domestication.

To start testing this hypothesis, we generated and compared de novo whole-transcriptomes from one representative population each of Cx. pipiens f. pipiens and f. molestus using the Cx. quinquefasciatus genome (CpipJ1.3 Johannesburg, South Africa, (Arensburger et al., 2010) as a reference. Cx. quinquefasciatus is a closely related sibling species of Cx. pipiens (Farajollahi et al., 2011), and is the only available annotated Culex genome assembly. We performed pairwise comparisons of orthologous coding (CDS) nucleotide sequences to identify genes and gene ontologies that show evidence of evolving at accelerated evolutionary rates between f. pipiens and f. molestus by calculating per-gene rates of non-synonymous substitution per non-synonymous site (Ka, or dN). Wang et al. (2011) show that commonly used tests for natural selection that normalize Ka by a ‘background mutation rate,’ or Ks (synonymous substitutions per synonymous site) often produce non-uniform results among closely related genomes, yet find that Ka alone remains stable and an adequate gauge for rate of “uncorrected” peptide evolution. This is primarily due to the varying manner in which Ks is calculated in a likelihood framework by different algorithms, and can also be influenced by sequence composition (Parmley & Hurst, 2007; Wang et al., 2011). Additionally, Ka/Ks calculations are often incorrectly elevated among isolated populations and closely related lineages due to segregating polymorphisms (both neutral and slightly deleterious) present at the time of divergence (Kryazhimskiy & Plotkin, 2008; Mugal, Wolf & Kaj, 2014; Peterson & Masel, 2009). Since there is minimal phylogenetic distance between the two forms we sequenced, synonymous substitutions would be expected to far outnumber those that are non-synonymous. This scenario is particularly susceptible to the aforementioned biases, as even small stochastic variation in synonymous substitution rates coupled with artifacts in Ks calculation can exert disproportionately large influence on the selection signature (Koonin & Rogozin, 2003; Parmley & Hurst, 2007; Wang et al., 2009). For these reasons, we elected to use Ka as the primary metric for presentation of our data. As the software we selected for our calculations implements the test in a likelihood framework which corrects for multiple substitutions at sites, a process less likely to have occurred in such closely related taxa, we performed primary calculations using also observed substitutions in addition to those derived from the model and discuss congruence between the two approaches. Although our primary objective was to elucidate components of the mosquito genome evolving at accelerated rate, we also report here ontologies enriched in the set of genes devoid of non-synonymous substitutions as they provide candidates for targets of negative or purifying selection and define critical biological processes and cellular components in the Cx. pipiens genome.

To contrast the amount of genetic variation uncovered in the comparison of Cx. pipiens forms with that of another geographically isolated yet conspecific population, we repeated the analysis with publicly available transcriptome data from two strains of Cx. quinquefasciatus: a North American strain (Reid et al., 2012) and the Johannesburg reference (Arensburger et al., 2010). We hypothesized that a greater amount of divergence would be witnessed between the two Cx. pipiens populations, which exhibit qualifiable phenotypic differences characteristic of the taxonomic forms, rather than between conspecific Cx. quinquefasciatus populations. In addition, we examined whether particular GO terms present in our results may be derived from ambiguous placement of read data from paralogous or multiple-copy genes by testing for their presence within an enriched ontology list derived from genes which share significant DNA similarity with others in the genome.

Materials and Methods

Because only Cx. pipiens f. molestus or hybrids of the two Cx. pipiens forms occur in the U.S, we obtained egg rafts of f. pipiens from Baden-Württemberg in southwestern Germany. Multiple individual egg rafts were isolated, hatched and DNA was extracted from ca. 10 larvae from each using a Qiagen DNEasy Blood & Tissue kit (Qiagen, Valencia CA). PCR-based positive species identification of Cx. pipiens was performed via the acetylcholinesterase-2 assay developed by Smith & Fonseca (2004), and further to f. pipiens using the CQ11 assay of Bahnck & Fonseca (2006). Field populations of pure f. molestus are difficult to obtain since they are strictly subterranean and mostly found by chance (Fonseca DM personal experience). Therefore, egg rafts of f. molestus were obtained from a young colony, initiated from a large subterranean swarm of females detected in a New York, NY residential basement in December 2010. Blooded females that had been biting local residents were allowed to lay egg rafts in the laboratory and henceforth the colony has been maintained without access to blood. Representative specimens of the NYC colony of f. molestus have been genotyped with a panel of 8 microsatellite loci and have a genetic signature that matches that of populations of f. molestus from southwestern Germany, as do other f. molestus specimens obtained from multiple locations around the world (Fonseca et al., 2004b; Micieli et al., 2013; Turell, Dohm & Fonseca, 2014). Once eggs hatched, larvae of both forms were reared in ceramic pans under a 16:8 L:D cycle on a diet of ground rat chow prior to emergence. Four specimen groups were created: thirty 1st/2nd instar, eight 3rd/4th instar, eight pupae and eight non-blood fed adult (4 male, 4 female) mosquitoes. Each group was placed in a separate plastic 2ml microcentrifuge tube containing a 5 mm sterile stainless steel bead and 900ul QIAzol lysis reagent prior to disruption with a TissueLyser II (Qiagen, Valencia, California, USA) for 2 min at 20 Hz. Total RNA extraction was then carried out on each group using the RNeasy Plus Universal kit (Qiagen, Valencia, California, USA) per manufacturer protocol and quantified on a Qubit 2.0 fluorometer (Life Technologies) using the RNA Broad-range buffer. One ug of RNA from each group was combined and used to prepare an Illumina sequencing library using the TruSeq RNA Sample Prep kit v2 (Illumina, Inc. San Diego, California, USA) per manufacturer protocol. The Cx. pipiens f. molestus library was sequenced twice on an Illumina MiSeq (Illumina, Inc., San Francisco, California, USA), once using a 500-cycle (2x250 bp paired-end) MiSeq Reagent Kit v2, and once using 1/3 of a multiplexed 600-cycle (2x300 bp paired-end) MiSeq Reagent Kit v3. Culex pipiens f. pipiens was sequenced once using 1/3 of a multiplexed 600-cycle (2x300 bp paired-end) MiSeq Reagent Kit v3. Raw sequence data were quality trimmed using the CLC Genomics Workbench (Limit score cutoff = 0.05, CLC Bio, Aarhus, DK).

To assemble EST sequences for each mosquito taxon (illustrated in Fig. 1), we used the sequenced genome of another recognized member of the Cx. pipiens complex, Culex quinquefasciatus Say (Arensburger et al., 2010) (for current taxonomy see http://wrbu.si.edu) as a reference. We mapped raw read data for each form individually to the Cx. quinquefasciatus genome CDS sequence, extracted from the CpipJ1.3 genome assembly available via VectorBase (http://www.vectobase.org/organisms/Culex-quinquefasciatus, (Megy et al., 2012)) using the CLC Genomics Workbench (CLC Bio, Aarhus, Denmark) at a nucleotide similarity of 95% over a required length fraction of 95% of the read. Reads that had more than one best alignment (i.e., potentially paralogous DNA) were ignored. Consensus sequences for each CDS were then generated from the alignment, with conflicts resolved by choosing the base with the highest additive quality score and a minimum coverage of 2x. Areas of <2x coverage were filled with Ns from the reference. The f. pipiens and f. molestus CDS sequences were aligned with each other, and sites with Ns in either or both forms were removed. Genewise (Birney et al., 2004) was used to create in-frame CDS sequences using the homologous peptide sequence of the Cx. quinquefasciatus as a guide, and any sequences that had stop codons introduced after this process were removed. Codon alignments were created with TranslatorX (Abascal, Zardoya & Telford, 2010), guided by a peptide alignment of their translations generated via MAFFT v.6.9 (Katoh & Toh, 2010). This codon alignment was used to calculate Ka values using the KaKs Calculator v.2 (Wang et al., 2010) using both observed non-synonymous substitutions and those estimated via maximum-likelihood estimation under likelihood model averaging (MA). We retained Ka values for CDS codon alignments greater than 200 bp, or for alignments <200 bp for which >50% of the sequence length (as calculated from the Cx. quinquefasciatus homolog) was recovered in the f. molestus—f. pipiens comparison. As this test compares single haploid gene sequences, and we reduced allelic variation within and among individuals sequenced from the population by generating haploid consensus gene sequences (above), it is likely that our Ka calculations underestimate the true amount of non-synonymous variation within the populations sequenced. Additionally, the alignment stringency (95%) of the mapping will exclude genes that have diverged significantly between the subject and the reference; however, we find it a conservative value with which to avoid false positives generated from gene paralogs. Enrichment tests were performed using Blast2GO (Conesa et al., 2005) with a reference set consisting of 11,930 genes (Table S1) that met the length criteria above (GO Term Filter Value =.05, Term Filter Mode = FDR, single-tailed test) and a test set composed of the 95th percentile of CDS sequences with highest calculated Ka. Additionally, to discern possible candidates of purifying selection, a test set of genes lacking non-synonymous substitutions from the f. pipiens—f. molestus comparison was created by selecting 4,575 CDS alignments (generated above, Table S1) from our data with 100% amino acid identity and used in a separate enrichment test coupled with the reference set above.

Figure 1 Illustration of codon alignment generation process.

(1) Illumina short read data are aligned to Cx. quinquefasciatus reference CDS sequence and used to build consensus sequences for both Cx. pipiens forms pipiens and molestus. (2) Consensus sequences for each gene are aligned, homologous positions free of Ns are removed and spliced. (3) GeneWise is used along with the corresponding full length Cx. quinq. peptide to create in-frame f. pipiens/f. molestus EST sequences from spliced alignments. (4) Codon alignments are created from EST sequences using TranslatorX. Ns denote unknown and/or unrecovered nucleotide data.

For the intra-specific Cx. quinquefasciatus comparison, data generated by Reid et al. (2012) from colonies started from an Alabama, USA population (strain HAmCq1 and HAmCq8) were compared to the CpipJ1.3 reference as above; briefly, reads from NCBI SRA libraries SRR364515 and SRR364516 were combined and mapped to the CpipJ1.3 CDS sequence, consensus sequences were built using the same protocol and parameters as above, and genewise/translatorX were used to construct the codon alignment prior to Ka calculation. From this, we constructed a reference set containing 13,281 genes which met the f. pipiens–f. molestus length cutoff above. As this was a conspecific comparison (assuming minimal evolution), we used only observed substitutions as opposed to those derived via maximum likelihood estimation (MLE) for the Ka calculation.

To examine whether particular gene ontologies present in our results may be derived via ambiguous placement of read data from paralogous or multiple-copy genes, we tested for their presence within an enriched ontology list derived from genes that share significant DNA similarity with others in the genome. This was accomplished by blasting the Cx. quinquefasciatus CpipJ1.3 CDS sequence data used above into itself via BLASTN (Altschul et al., 1990) with an e-value cutoff of 1 × 10−5 and saving all ‘non-self’ hits for genes which had a 95% similarity over a local alignment of 200 nt (a value we chose as our average read length after trim was 211 nt). This returned 3,687 (Table S11) sequences that were used as a test set in a Blast2GO enrichment test against a reference consisting of all CDS sequences.

In all tests, we retained GO terms with a False Discovery Rate (FDR) corrected (Benjamini & Hochberg, 1995) p-value of p ≤ .05. Gene names reported are retained from the Cx. quinquefasciatus reference used to construct the consensus. Annotations were performed against the NCBI nr database and via InterProScan v.5 (Apweiler et al., 2000). Phylogenetic analysis of the Peritrophin-A domain-containing proteins was performed by extracting the peptide sequence for each chitin-binding domain from the Cx. quinquefasciatus homolog corresponding to each of our candidate genes based on coordinates returned via InterproScan v.5 prior to alignment with a selection of peritrophic matrix protein (PMP) and cuticular proteins analogous to peritrophin (CPAP) domains of Jasrapuria et al. (2010) extracted in the same manner. Sequences were aligned using T-COFFEE v.10.00.r1613 (Notredame, Higgins & Heringa, 2000) and tree reconstruction under automatic model selection and 1500 bootstrap replicates was performed using IQTREE v. 0.9.6 (Minh, Nguyen & von Haeseler, 2013).

Results and Discussion

Transcriptome sequencing and Ka calculation

Transcriptome sequencing generated 58.7 million (11.2 Gbp) and 24.7 million (5.3 Gbp) of short-read data for f. molestus and f. pipiens, respectively. The f. molestus data mapped to 18.4 Mbp (74%) of the 25.0 Mbp Cx. quinquefasciatus CDS sequence reference by length (15,624 of 19,019 transcripts had at least one mapped read), with an average coverage of 71x and median coverage (50th percentile) value of 17x. The f. pipiens RNAseq data mapped to 17.2 Mbp (70%) of the Cx. quinquefasciatus reference by length (14,537 transcripts had at least 1 mapped read) with an average coverage of 45.5x and median of 8x at our alignment stringency (95% nt similarity over 95% of the read length, see Methods). After refinement by length and coverage (see Methods), the short read alignments were used to create 11,930 pairs of putative ortholog consensus sequences (one pair for each of 11,930 genes). Each taxon contributed 14.15 Mbp of sequence data. After codon alignment, the gene set was ranked by pairwise Ka value calculated via both the maximum-likelihood estimation and by observed count, and the top 5% (n = 597, Table S1) of genes from each were selected to create two Blast2GO test sets for Enrichment Analysis (Fisher’s Exact Test).

Enrichment within the fast-evolving genes

When reduced to most-specific terms (i.e., parent terms removed), the analysis identified the same seven Gene Ontology (GO) terms as enriched for both the observed and log-likelihood test sets (Table 1): serine-type endopeptidase activity (GO0004252), proteolysis (GO0006508), receptor binding (GO0005102), odorant binding (GO0005549), extracellular space (GO0005615), chitin metabolic process (GO0006030) and chitin binding (GO0008061). As both test sets converged on the same terms, we will present all further results and data tables corresponding to output from the observed count analysis.

Table 1 GO terms enriched in fast-evolving genes.

Gene ontology terms enriched in the upper 95th percentile of pairwise dN values calculated using Culex pipiens forms pipiens and molestus homologous codon sequence alignments.

GO ID	Go term	FDR	p	# in test
set	# in ref.
set	# unannotated
test set	# unannotated
reference set	
GO:0004252	Serine-type endopeptidase activity	1.20E−13	7.60E−17	51	232	364	7988	
GO:0006508	Proteolysis	1.40E−09	1.80E−12	71	546	344	7674	
GO:0005102	Receptor binding	7.50E−09	1.50E−11	25	80	390	8140	
GO:0005549	Odorant binding	1.40E−06	3.20E−09	16	39	399	8181	
GO:0005615	Extracellular space	7.30E−04	2.00E−06	10	23	405	8197	
GO:0006030	Chitin metabolic process	5.80E−03	1.70E−05	17	93	398	8127	
GO:0008061	Chitin binding	1.20E−02	4.80E−05	15	81	400	8139	

The Serine-type endopeptidase activity (GO:0004252) ontology comprises a family of enzymes that utilize a nucleophilic serine at the active site to cleave peptide bonds in proteins. These enzymes are widely distributed throughout both pro- and eukaryotes and classified into 16 superfamilies. Most eukaryotic serine endopeptidases belong to the Chymotrypsin serine protease S1 family, where both chymotrypsin-like and trypsin-like proteases function as digestive enzymes in hydrolyzing proteins to smaller peptides and amino acids for further digestion (Madala et al., 2010; Rawlings & Barrett, 1994). Annotation of the serine endopeptidases within our enriched set (Table S2) shows 45 of the 50 proteins carry a trypsin domain (Pfam PF00089). Mosquito trypsins, secreted by gut epithelium, function in digestion of protein-rich bloodmeals within the female after encapsulation by a peritrophic matrix (Borovsky, 2003; Borovsky & Schlein, 1987). In a process currently considered unique to mosquitoes (Diptera: Culicidae), two forms of trypsin are critical for complete bloodmeal digestion (Felix et al., 1991). Within 1 h following ingestion, early trypsin protein is translated from mRNA stored in the gut epithelium. This early trypsin protein functions to partially digest the bloodmeal, creating smaller peptides that in turn trigger and regulate late trypsin transcription and translation (Borovsky, 2003; Noriega, Colonna & Wells, 1999). Late trypsins then further digest the bloodmeal to free amino acids sourced for egg development. This feedback mechanism ensures that digestive proteases are produced only in response to blood (as opposed to carbohydrate/sugar) and in quantities commensurate with “pre-assessment” of bloodmeal protein content by early trypsin digestion. In addition to digestion, Valenzuela et al. (2002) found several secreted salivary serine proteases with homology to Manduca prophenoloxidase-activating enzymes that are likely involved in the innate melanotic immune response.

The presence of such elevated levels of trypsin variation between populations may indicate that differences in the source of bloodmeal necessitated adaptive changes in digestive enzymes to hydrolyze differentially abundant proteins. Further study will be required to determine whether the proteins highlighted in our analysis represent early and/or late trypsins, as two proteins carried an annotation of late trypsin and only four trypsins have been annotated as early or late to date within the Cx. quinquefasciatus genome project (via Vectorbase; https://www.vectorbase.org/organisms/culex-quinquefasciatus, retrieved Jun 2014). Five proteins in our set were annotated as coagulation factors; however, an NCBI Conserved Domain analysis (http://www.ncbi.nlm.nih.gov/Structure/cdd/wrpsb.cgi, results not shown) fails to return evidence for canonical Gla and/or EGF domains within these peptides, indicative of the coagulation factors (Stavrou & Schmaier, 2010).

The proteolytic enzymes within the Proteolysis (GO:0006508) ontology hydrolyze proteins to smaller peptides and/or amino acids. This gene ontology contained primarily the serine endopeptidase enzymes discussed above, with the addition of several serine protease inhibitors, metallopeptidases and apoptotic caspases (Table S3).

Receptor binding (GO:0005102) protein molecules interact selectively with specific cellular receptors to initiate changes in cell function. Eighteen such proteins were present in the enriched set, of which all were found to carry a fibrinogen beta and gamma chain Pfam (PF00147, Table S4) annotation. In the invertebrates, including mosquitoes, fibrinogen-related proteins (FREPs) are restricted to the innate immune response, functioning in pathogen recognition and agglutination (Dong & Dimopoulos, 2009; Hanington & Zhang, 2011). Many Anopheles gambiae FREP genes display immune-responsive transcription after being challenged with bacteria, fungi or both rodent and human malaria protozoa (Dong & Dimopoulos, 2009) indicating that they play a pivotal role in mosquito vectorial capacity. This gene family has undergone lineage-specific duplications with relaxed selective constraints, as the An. gambiae genome contains 59 FREP members, with 32 and 87 members currently annotated in the genomes of Ae. aegypti and Cx. quinquefasciatus, respectively (Arensburger et al., 2010), while the Drosophila melanogaster genome contains twenty (Wang, Zhao & Christensen, 2005). This likely reflects the diverse pathogen load faced by each particular dipteran species during its life cycle. Further annotation reveals four putative ficolins in our set, a particular oligomeric lectin containing a C-terminal fibrinogen-like domain able to bind N-acetylglucosamine, a chitin monomer, as part of immune response (Krarup et al., 2004). It is likely that the two populations of Cx. pipiens sequenced here are challenged by different bacterial communities within their respective environments, and experience both varying larval habitat (subterranean sewers and subway systems [form molestus] vs. stagnant, above-ground pools [form pipiens]) and bloodmeal hosts (with associated food-borne pathogens; see Serine endopeptidases above). The rate of peptide evolution seen in this component of the innate immune system may be a result of adaptation to these ecological stressors.

Members of the odorant binding (GO:0005549) ontology compose a large multi-gene family of water-soluble proteins secreted by support cells into sensillum lymph of the female mosquito antennal hairs (Schultze et al., 2013). These proteins bind various odorant molecules, thus triggering chemosensory mechanisms such as host-seeking and oviposition site recognition (Pelosi & Maĭda, 1995). Characterized by a six alpha-helical domain and the disulphide bonds created by six conserved cysteine residues, the mosquito odorant binding proteins (OBPs) have been studied extensively in the available mosquito genomes. Like the fibrinogens, the OBP protein family has been found to be very divergent within the Culicidae, with low sequence identity between interspecific homologs (Vieira & Rozas, 2011) and can be further divided into four subfamilies: (1) Classic OBPs, which conform to the domain characterization above, (2) PlusC and MinusC OBPs, which contain six additional disulfide-bonded cysteine residues or lack two, respectively (Hekmat-Scafe et al., 2002), and (3) Atypical OBPs, which contain two complete Classic OBP domains (e.g., “dimer OBPs”, (Vieira & Rozas, 2011)). In a recent study, Manoharan et al. (2013) expanded the number of known OBPs from the three published mosquito genomes by 110 members to a total of 289, while classifying each by subfamily. Ascribing function to peptides based on sequence homology to known OBPs can prove difficult. Leal (2005) note that several gene families with OBP-like domain structure show no evidence of involvement in olfactory or pheromone-mediated responses, and suggests the term “encapsulins” supersede “odorant-binding proteins” to more accurately describe the common function (ligand encapsulation) performed by the peptide.

An additional protein family often included in evolutionary analyses of mosquito OBPs is the D7 salivary protein family, which exhibits domain structure similar to that of the OBPs with the addition of a seventh helix (Kalume et al., 2005). Classified into short (15–20 kDa) and long (30–36 kDa) subfamilies, the long-form D7 salivary proteins contain a second OBP-like domain in an N-terminal extension (Calvo et al., 2006; Valenzuela et al., 2002). The singular domain in the short-form and C-terminus of the long-form salivary D7 protein has been shown to bind biogenic amines (serotonin, histamines and norepinephrine) with high affinity, while the N-terminal domain of the long-form protein binds leukotriene inflammatory mediators, thus inhibiting platelet aggregation, vasoconstriction and inflammation (collectively hemostasis) during blood-feeding (Calvo et al., 2006; Calvo et al., 2009a).

Our analysis identified sixteen proteins with an odorant binding cellular function (Table S5), of which fourteen carried a Pfam ID of PF01395 (PBP/GOBP Family). Annotation of these proteins via Vectorbase reveals the list is comprised of six D7 salivary peptides, representing 60% of the known D7 proteins in the Cx. quinquefasciatus genome (n = 10, https://www.vectorbase.org/organisms/culex-quinquefasciatus) and eight odorant-binding proteins. The Cx. quinquefasciatus homologs of all OBPs in our set were recently classified by Manoharan et al. (2013), which allowed us to further assign our representatives to subfamily and cluster. Seven of the eight proteins were of the Classic OBP subfamily, i.e., containing a singular OBP domain, with four of these being minus-C type and lacking two of the canonical cysteines.

These results indicate that the transcriptome of the two representative Cx. pipiens populations sequenced were most divergent within their odorant-binding domain-containing proteome at the D7 salivary proteins, and predominantly among the minus-C forms of the Classic Odorant-binding protein subfamily. Since the two forms differ in their propensity for taking mammalian (including human) vs. bird bloodmeals (Huang, Molaei & Andreadis, 2008; Osório et al., 2014) the particular OBP subset highlighted here may contribute to the olfactory response to differing host cues. Additionally, the oviposition habitat available to subterranean mosquitoes (i.e., sewers) likely presents olfactory cues that differ from those above ground. The concomitant chemosensory response may necessitate evolution of OBP-encoding genes. As all but one OBP in our set were newly described by Manoharan et al. (2013) and were not included in the tissue-specific expression analysis of Leal et al. (2013), it is unknown whether they may be localized to antennae, palps or other somatic tissues. However, the representation of D7 salivary proteins in the enriched set may indicate that the immunosuppressive complement of mosquito saliva has diverged in accordance with local environment. The mosquito sialome has previously been shown to exhibit accelerated evolutionary pressures at the interspecific level; in a comparative analysis of New World (An. darlingi) and Old World (An. gambiae) Anopheline sialotranscriptomes, Calvo et al. (2009b) found that on average, salivary proteins were only 53% identical at the amino acid level as opposed to 86% identity among housekeeping genes.

Components of the extracellular space (GO:0005615) gene ontology exist outside the cell plasma membrane within interstitial fluids. Our test set contained ten such proteins (Table S6), with seven fibrinogens discussed above (and annotated as having extracellular localization) being re-listed here. The remaining three proteins were of the macroglobulin complement family, which carry alpha-2 macroglobulin family N-terminal (Pfam PF07703) and alpha-macroglobulin receptor (Pfam PF07677) domains. Alpha-2 macroglobulins (α2M) are proteinase-binding and inhibiting glycoproteins commonly secreted by hemocytes within insect hemolymph (Sottrup-Jensen et al., 1989), which have been found recently to play integral roles in complement-like pathways that bind parasite surface targets (Blandin, Marois & Levashina, 2008). The full-length protein exposes a “baited” peptide stretch, which when cleaved by proteinases present with septic injury will change protein conformation to an active state that covalently binds the activating proteinase. This conformational change also exposes binding sites with high affinity for both gram-positive and negative bacteria (Blandin, Marois & Levashina, 2008; Sottrup-Jensen et al., 1989). The complex is then targeted for phagocytosis. Like the fibrinogens, the presence of these proteins in the most diverged set indicates that the two populations of Cx. pipiens may experience very different microbiome challenges, consistent with the differences between forms (e.g., utilization of sewers) in larval habitat (Harbach, Harrison & Gad, 1984). Furthermore, as α2M inhibits the coagulation proteinases thrombin and factor Xa, it serves to inhibit the coagulation cascade and thus may function in blood-feeding hemostasis as well (de Boer et al., 1993).

The Chitin metabolic process (GO:0006030) ontology (inclusive of all genes composing the Chitin binding (GO:0008061) ontology, Tables S7 and S8) is composed of reactions and pathways involving chitin, a linear polysaccharide polymer that consists of linked glucosamine residues and forms the main component of arthropod exoskeleton, tracheae and peritrophic membrane (PM). Seventeen proteins in the test set were annotated with this term; eleven with a Pfam Chitin binding Peritrophin-A domain (PF01607). The additional two peptides were annotated with a chitinase molecular function, each with two Pfam Chitinase class I domains (PF00182). Peritrophins are structural proteins consisting of one to many chitin-binding domains responsible for cross-linking chitin fibrils (Wang & Granados, 2001). The semi-permeable lattice created, known as the peritrophic membrane, surrounds the insect food bolus and separates it from the midgut epithelial cells. This serves to protect the gut (and insect) from physical damage, pathogens and toxins. There is evidence that the PM plays a central role in binding toxic free heme via the chitin-binding domain (CBD) (Devenport et al., 2006; Pascoa et al., 2002) during bloodmeal digestion, indicating free CBDs on bound peritrophins of the PM serve additional purposes. Pascoa et al. (2002) found an amount of free heme bound to the Aedes aegypti PM equivalent to hydrolysis of 2ul of a typical 3ul bloodmeal. To determine whether our peptides were in fact peritrophins associated with a midgut PM, as opposed to non-specific cuticular proteins analogous to peritrophins (CPAPs, see Jasrapuria et al. (2010)) which also exhibit Peritrophin-A domain homologs, we aligned our nine candidate peptide domains to a selection of those from the classification of Jasrapuria et al. (2010) and produced a maximum-likelihood tree which grouped all 21 of our sequences in a Peritrophic Matrix Protein (PMP) clade at a bootstrap support of 99% (Fig. S1). This indicates our candidates are in fact likely associated with the midgut PM and involved in digestion.

Chitinases are integral enzymes in the creation and destruction of the adult mosquito PM. Initially synthesized as a zymogen upon ingestion of a blood meal, it is later activated by removal of a propeptide from the N-terminus (Bhatnagar et al., 2003) and begins to hydrolyze the glycosidic linkages of the PM chitin matrix to chitobiose (a glucosamine dimer) as the blood meal is digested. Like the PM itself, chitinase enzymes are important research targets for pathogen defense. The Plasmodium parasite ookinete expresses a mosquito chitinase ortholog able to accelerate PM degradation and facilitate escape (Langer & Vinetz, 2001) Bhatnagar et al. (2003) were able to utilize the inhibitory effect of the propeptide on its cognate enzyme to block chitinase activity in both Anopheles gambiae and Ae. aegypti, thus suppressing development of human and avian Plasmodium, respectively, in the two mosquito species. Initial blood meal digestion within the female midgut requires transit of trypsins across the PM, and later, diffusion back to the ectoperitrophic space (Terra & Ferreira, 1981). The peritrophic membrane has important dual-responsibilities in digestion and immunity, two systems we have associated with other enriched GO terms, further implicating this structure as a driving force in the differentiation of the two Cx. pipiens populations.

The insect immune system has been shown to be a common target of positive selection (Bulmer, 2010; Roux et al., 2014), and the role it plays in differentiation of these two mosquitoes is further exemplified by examining the gene with the largest calculated Ka in our comparison (Table S1), a homolog of CPIJ006559 representing a peptidoglycan recognition protein (PGRP) containing a N-acetylmuramoyl-L-alanine amidase (Pfam PF01510). This particular PGRP (PGRP-LC) is a transmembrane molecule that, upon binding bacterial peptidoglycan, triggers the immune deficiency (Imd) pathway in Drosophila (Choe, Lee & Anderson, 2005). A manual scan of our test set for other immune-related peptides that may bind peptidoglycan and/or carbohydrate yields eight proteins with a carbohydrate binding cellular function (GO:0030246) of which seven are lectins, with 5 annotated as salivary C-type lectins. These likely serve in food-borne pathogen identification (Ribeiro et al., 2004; Valenzuela et al., 2002); however, the possibility exists that these proteins function instead as anti-clotting agents as has been reported in snake venom (Koo et al., 2002) and in the phlebotomine sand fly Lutzomyia longipalpis (Charlab et al., 1999).

Highly conserved genes

An enrichment test using the gene set devoid of non-synonymous substitutions from the f. pipiens—f. molestus comparison retained 19 significantly enriched GO terms (Table 2). These included primarily transcription and translational machinery (Structural constituent of ribosome, rRNA binding, Transcription regulatory region sequence-specific DNA binding), cell signaling components (GTP binding, GTPase mediated signal transduction, postsynaptic membrane, cell junction, G-protein coupled receptor signaling, outer membrane-bound periplasmic space, MAPK cascade, regulation of ion transmembrane transport) and ATP coupled proton transport (ATP hydrolysis coupled proton transport, ATP synthesis coupled proton transport, proton-transporting V-type ATPase). Of particular interest were the four GO terms for which all members were present in the enriched set only (i.e., the GO term constituents contained only synonymous substitutions; Table S13): (1) outer-membrane bound periplasmic space (GO0030288) contained glutamate receptors responsible for postsynaptic excitation of insect neuronal and muscle cells (Briley et al., 1981), (2) the MAPK cascade (GO0000165) that communicates biotic and abiotic signals from extracellular ligands to the nucleus, initiating a response (e.g., division, apoptosis, etc.) from the cell (McKay & Morrison, 2007), (3) proton-transporting V-type ATPases (GO0033180) that are a diverse and highly conserved membrane-spanning enzyme coupling ATP hydrolysis to proton transport (Nelson et al., 2000), and (4) the transcription regulatory region sequence-specific DNA binding ontology (GO0000976) that contains several homeobox domains encoding transcription factors which activate and regulate patterns of morphogenesis (Gehring, 1992). Several of these pathways have been previously described as highly conserved in eukaryotes (Bejerano et al., 2004; Li, Liu & Zhang, 2011), and when taken together define a genetic “core” in Cx. pipiens that confer critical phenotypes and cellular processes refractory to amino acid substitutions and are the strongest candidates for negative or purifying selective pressures.

Table 2 GO terms enriched in slow-evolving genes.

Gene ontology terms enriched in the set of 4,575 pairwise Culex pipiens forms pipiens and molestus homologous codon alignments devoid of non-synonymous substitutions.

GO ID	Go term	FDR	p	# in test
set	# in ref.
set	# unannot.
test set	# unannot.
reference set	
GO:0003735	Structural constituent of ribosome	1.00E−15	7.60E−19	98	30	3209	5298	
GO:0005525	GTP binding	2.70E−09	1.70E−11	113	67	3194	5261	
GO:0007264	Small GTPase mediated signal transduction	2.90E−04	6.40E−06	104	89	3203	5239	
GO:0051301	Cell division	2.50E−03	8.60E−05	27	12	3280	5316	
GO:0007186	G-protein coupled receptor signaling pathway	4.10E−03	1.50E−04	76	66	3231	5262	
GO:0003924	GTPase activity	6.60E−03	2.50E−04	54	42	3253	5286	
GO:0030288	Outer membrane-bounded periplasmic space*	1.10E−02	4.60E−04	8	0	3299	5328	
GO:0030054	Cell junction	1.30E−02	5.70E−04	28	16	3279	5312	
GO:0004930	G-protein coupled receptor activity	1.80E−02	8.10E−04	45	35	3262	5293	
GO:0000165	MAPK cascade*	2.50E−02	1.20E−03	7	0	3300	5328	
GO:0006334	Nucleosome assembly	2.70E−02	1.40E−03	18	8	3289	5320	
GO:0005509	Calcium ion binding	3.10E−02	1.60E−03	103	110	3204	5218	
GO:0015991	ATP hydrolysis coupled proton transport	3.50E−02	1.90E−03	14	5	3293	5323	
GO:0019843	rRNA binding	3.50E−02	1.90E−03	10	2	3297	5326	
GO:0015986	ATP synthesis coupled proton transport	3.50E−02	1.90E−03	10	2	3297	5326	
GO:0034765	Regulation of ion transmembrane transport	3.80E−02	2.10E−03	15	6	3292	5322	
GO:0045211	Postsynaptic membrane	4.50E−02	2.70E−03	19	10	3288	5318	
GO:0033180	Proton-transporting V-type ATPase, V1 domain*	5.00E−02	3.10E−03	6	0	3301	5328	
GO:0000976	Transcription regulatory region sequence-specific DNA binding*	5.00E−02	3.10E−03	6	0	3301	5328	
Notes.

* indicate terms for which all members were present only in the test set.

Comparison between geographically isolated populations

The populations of Cx. pipiens forms pipiens and molestus mosquitoes sequenced in this study were geographically isolated. To assess how the amount of variation between Cx. pipiens forms (defined by number and IDs of enriched GO terms) reported in our analyses compared to conspecific isolated populations, we repeated our analysis using publicly available data from a recently colonized population of Cx. quinquefasciatus isolated from Alabama, USA (Reid et al., 2012) and the CpipJ1.3 Johannesburg reference CDS sequences. Short-read mapping produced alignments covering 17,410 of 19,019 CDS sequences with >1 read and covered 19.8 Mbp (79%) of the reference, with average coverage of 133x and median coverage of 14.7x. After applying the length and 2x coverage cutoff (see Methods), we retained 13,586 CDS codon alignments for analysis with the 95th percentile test set composed of 679 sequences (Table S9). Blast2GO analysis retained only two significant GO terms when reduced to the most-specific set (Table S10). Neither of these terms (both composed of the same seven genes encoding reverse transcriptase enzymes and retrotransposons) are present in our Cx. pipiens comparison, indicating that the f. pipiens–f. molestus populations sampled here maintain a greater degree of evolutionary protein divergence than the isolated yet conspecific Cx. quinquefasciatus populations.

Assessing effects of paralogy and sequence identity

Some of the gene families and protein domains reported in this study are among the most abundant in the mosquito genome. For example, Interproscan5 analysis of the CpipJ1.3 transcripts (not shown) uncovers 477 trypsin and 283 peritrophin-A domains. Even though we discarded sequencing reads with multiple top-scoring genome alignments, to ensure our results were not reflective of incorrect short read placement among multiple paralogous genes, we tested the propensity for our reported GO terms to be enriched among those genes that share significant sequence identity to others in the genome. Using all CpipJ1.3 CDS sequences with BLASTN alignments ≥200 bp at ≥95% similarity to another CDS in the genome (Table S11), we derived a test set which contained 41 enriched terms (Table S12). This list contained no GO terms previously reported here, thus we find no evidence that the resultant terms from our f. pipiens–f. molestus comparison originate from gene families with biased sequence identity.

Conclusions

These are the first insights into the genome-wide molecular differentiation of two closely related yet phenotypically divergent populations of an important disease vector, Cx. pipiens. Analysis of over-represented gene ontology terms within the fastest evolving peptides elucidates the biological systems that are targets of local adaptation. Although further analyses with additional representative populations of the two forms are necessary, our results likely hold clues as to the molecular mechanisms responsible for phenotypic divergence between the two taxonomic forms, and subsequently confer Culex pipiens form molestus the ability to exploit human environments. The recurring localizations within our data to gene families functioning in odorant binding, hemostasis, digestion, and innate immunity can all be linked to a differential propensity of these forms to seek a mammalian host, ability to obtain and process a bloodmeal, and to thrive as larvae and adults in subterranean sewers rich with organic wastes and associated bacteria. In addition, we provide candidate loci for future functional in-vivo assays to qualify effects on phenotype. Interestingly, of the seven GO terms reported in this study, five terms (chitin metabolic process, chitin binding, serine-type endopeptidase activity, proteolysis and odorant binding) were enriched along the ‘fly’ branch (represented by the Drosophila melanogaster genome (Adams et al., 2000)) of the branch-site selection tests conducted by Roux et al. (2014), indicating they may represent a genetic ‘core’ remaining under selection and responsible for adaptive evolution within the Diptera. Further sequencing of members of the Culex pipiens complex (Farajollahi et al., 2011) will enable additional tests involving lineage-specific estimates of evolutionary rates (e.g., Mensch et al., 2013) and definition of functional classes of genes with significantly elevated selection coefficients as compared to ancestral states in the phylogeny (Serra et al., 2011), as well as defining the role of differential gene expression in the divergence of a global mosquito vector.

Supplemental Information

Figure S1 Peritrophin-A phylogenetic tree

Maximum-likelihood phylogenetic tree showing monophyly of peritrophin-A domains reported here with peritrophic matrix proteins (labeled PMP), exclusive of the cuticular proteins analogous to peritrophins (labeled CPAP) of Jasrapuria et al. (2010). NCBI GI numbers are appended to Tribolium castaneum sequence IDs; all sequences are suffixed with “_subseq_[coordinate of first amino acid extracted]-[length of extracted peptide window]”.

Click here for additional data file.

Table S1 Ka calculations, annotations and Pfam IDs for protein homologs in this study

Observed and estimated Ka calculations, annotation and top-scoring Pfam IDs corresponding with 11,931 pairwise Culex pipiens forms pipiens and molestus homologous codon sequence alignments (ordered by decreasing Ka). Columns two and three denote genes present in the 95th percentile as ranked by Ka calculated using observed and likelihood estimated non-synonymous substitutions, respectively.

Click here for additional data file.

Table S2 Gene set composing the serine-type endopeptidase ontology

Gene set composing the serine-type endopeptidase ontology, found to be enriched in the 95th percentile of top-scoring Culex pipiens forms pipiens and molestus homologous codon sequence alignments as ranked by Ka value.

Click here for additional data file.

Table S3 Gene set composing the proteolysis ontology

Gene set composing the proteolysis ontology, found to be enriched in the 95th percentile of top-scoring Culex pipiens forms pipiens and molestus homologous codon sequence alignments as ranked by Ka value.

Click here for additional data file.

Table S4 Gene set composing the receptor binding ontology

Gene set composing the receptor binding ontology, found to be enriched in the 95th percentile of top-scoring Culex pipiens forms pipiens and molestus homologous codon sequence alignments as ranked by Ka value.

Click here for additional data file.

Table S5 Gene set composing the odorant binding ontology

Gene set composing the odorant binding ontology, found to be enriched in the 95th percentile of top-scoring Culex pipiens forms pipiens and molestus homologous codon sequence alignments as ranked by Ka value.

Click here for additional data file.

Table S6 Gene set composing the extracellular space ontology

Gene set composing the extracellular space ontology, found to be enriched in the 95th percentile of top-scoring Culex pipiens forms pipiens and molestus homologous codon sequence alignments as ranked by Ka value.

Click here for additional data file.

Table S7 Gene set composing the chitin binding ontology

Gene set composing the chitin binding ontology, found to be enriched in the 95th percentile of top-scoring Culex pipiens forms pipiens and molestus homologous codon sequence alignments as ranked by Ka value.

Click here for additional data file.

Table S8 Gene set composing the chitin metabolic process ontology

Gene set composing the chitin metabolic process ontology, found to be enriched in the 95th percentile of top-scoring Culex pipiens forms pipiens and molestus homologous codon sequence alignments as ranked by Ka value.

Click here for additional data file.

Table S9 Ka calculations for Cx. quinquefasciatus instraspecific comparison

Ka calculations corresponding with 13,587 pairwise Culex quinquefasciatus strain HAmCq and CpipJ1.3 homologous codon sequence alignments. Column two denotes genes present in the 95th percentile as ranked by Ka calculated using observed non-synonymous substitutions.

Click here for additional data file.

Table S10 GO terms enriched in Cx. quinquefasciatus intraspecific comparison

Gene ontology terms enriched in the upper 95th percentile of pairwise dN values calculated using Culex quinquefasciatus strains HAmCq and CpipJ1.3 homologous codon sequence alignments.

Click here for additional data file.

Table S11 Cx. quinquefasciatus self-blast output

BLASTN output detailing the 3,687 Culex quinquefasciatus CDS sequences with at least one BLASTN alignment ≥200 bp at ≥95% similarity to another CDS in the genome.

Click here for additional data file.

Table S12 GO terms enriched in Cx. quinquefasciatus self-blast output

Gene ontology terms enriched in the set of 3,687 Culex quinquefasciatus CDS sequences with at least one BLASTN alignment >200 bp at >95% homology to another CDS in the genome.

Click here for additional data file.

Table S13 Ka calculations, annotations and Pfam IDs for slowest-evolving genes

Extended analysis for all genes belonging to the GO terms from the slowest-evolving set (Table 2) for which all members were present only in the test set, and contained only synonymous substitutions.

Click here for additional data file.

Supplemental File S1 Codon alignments generated in this study

Codon alignments generated in this study.

Click here for additional data file.

We are grateful to Linda McCuiston for her unsurpassed expertise in rearing and colonizing the mosquitoes used in our study, to Nicole Wagner at the Rutgers University School of Environmental and Biological Sciences Genome Cooperative for performing our Illumina Sequencing, and to Peter Armbruster for comments on the manuscript.

Additional Information and Declarations

Competing Interests

Author Contributions

DNA Deposition

Data Deposition

Dina M. Fonseca is an Academic Editor for PeerJ.

Dana C. Price conceived and designed the experiments, performed the experiments, analyzed the data, wrote the paper, prepared figures and/or tables, reviewed drafts of the paper.

Dina M. Fonseca conceived and designed the experiments, contributed reagents/materials/analysis tools, wrote the paper, reviewed drafts of the paper.

The following information was supplied regarding the deposition of DNA sequences:

Sequencing libraries have been submitted to the NCBI SRA archive and can be accessed via BioProject PRJNA275017:

http://www.ncbi.nlm.nih.gov/bioproject/275017.

The following information was supplied regarding the deposition of related data:

Sequence alignments have been provided as a supplemental file to the manuscript.

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
