# Peer review of "Genetic divergence between populations of feral and domestic forms of a mosquito disease vector assessed by transcriptomics"

_PeerJ, doi:10.7717/peerj.807_

## Round 0.1 · original submission · Major Revisions

This study is useful as a first approach comparing the two mosquito forms and for identifying candidate gene sets characterized by rapid divergence.

However, the authors are trying to extrapolate differences between two samples (one per mosquito form) to differences between forms. Obviously, more samples would be needed to identify gene families that could contribute to the phenotypic divergence between forms, especially considering that one of the two samples stems from a lab strain of mosquitos and the other one is a field sample (though the methods are not entirely clear about this). Thus, we can only consider a manuscript where conclusions regarding divergence between forms are toned down significantly (or removed where appropriate).

Reviewer 1 ·

Basic reporting

This is a well-written MS and is easily comprehensible by anyone in the field. In addition, the writing is such that those not completely involved in the field of insect genomics will be able to understand portions relevant to their work. The introduction/literature review is sufficient and covers the bases adequately.

Figures S1 and S2 might be fine in the main text instead of in supplementary material.

The results here are useful both in terms of basic transcriptomic information/data for a medically important insect, and in terms of providing a first glimpse of which particular genes may be viable targets for understanding this insect and for potential management/control. The MS represents a substantial amount of effort an analysis and stands well on its own as a complete study. It is bound to be relevant not only to the mosquito and Diptera research communities, but also to others working in insect functional genomics.

Experimental design

The M&M section reflects the underlying experimental design and provides sufficient detail as to be completely replicable. The methods and analyses are robust and rigorous and the discussion reflects the results.

Validity of the findings

As noted above, the results are robust and the statistical methods are stringent and appropriate. I make expanded mention of the stringency of the statistical methods in my "General Comments for the Author" as well.

One of the potential concerns – that of the comparison groups coming from distant populations – was dealt with well and was discussed in good detail by the authors.

As noted above, this work is an excellent addition to the field of mosquito biology and of insect functional genomics in general. I suspect that the data that have been generated will be used in a variety novel ways by other entomologists and zoologists in general.

Additional comments

Here are some more specific comments on the MS. Some of them recount what I've written above, and some of them expand on previous comments.

**Abstract:**
Clearly written and reflects the entirety of the MS. No concerns.

**Introduction:**

Lines 35-39: This sentence is grammatically fine, but is a bit confusing. Breaking it into separate sentences might be better.

Line 64: no need for quotation marks on “genetic basis”.

Line 65: “de-novo” is better written without the hyphen, and Latin terms are often set in italics.

The introductory material covers the topic in sufficient detail and sets the reader up for the rest of the paper. No further comments here.


**Methods and Materials:**

Lines 135-140: “Each group of specimens was placed…” What constitutes a group here? Not clear. From later text, they seem to have been grouped in life stages and (in the case of adults) sex? And what was extracted? Presumably the entire group?

Use of different life stages increases the likely diversity of transcripts that were available for observation.

Stringency was set sufficiently high to avoid Type I errors. Some possibility of Type II errors. But in my experience with genomics work, the amount of information gleaned is generally astounding even at high stringency levels. I.e., while there is a risk of Type II error, I think that the risk of Type I error is higher in the context of the amount of useful data that can be acquired. So the author’s choices seem apt in this case.

The Methods and Materials section gives enough detail for others to replicate this work.

I note that data has been deposited at Lab Archives. I’m not aware of these services. Are the data sufficiently archived/backed-up here so that they will be available in perpetuity? Are they given DOI designation? If not, then I suggest placing data with a service like figshare or Dryad. NCBI deposition for the Illumina libraries, as noted, is great.

**Results and Discussion:**

This section is very long, but that is sometimes unavoidable in studies like this. Some shortening in terms of explanation of generalities of several of the groups of genes/transcripts may be possible.

Line 259: quotes around “most-specific terms” do not need to be there.

Line 303, Line 306, 385, etc.: I am a bit concerned about the use of URLs in the text, as URLs have a way of changing or disappearing. In other words, readers some years from now may not have access to these. Are there better ways to provide this information?

Lines 350-357 and lines 512-526: Very complex lists, perhaps better expressed as several sentences or a numbered list within the text [i.e. (1), (2), etc.].

Section starting at Line 531: This is good be because it answers a very obvious question that crops up while reading the methods. Ditto for section starting at Line 552.

Line 583-584: Use square outside parentheses.

Reviewer 2 ·

Basic reporting

This article meets the PeerJ standards for 'Basic Reporting'.

Experimental design

This article does not meet the PeerJ standards for 'Experimental Design'. Specifically, the investigation has not been rigorously conducted to a high technical standard. The technical problem with this MS is that it only compares one pair of populations. This sample size of one is really uninformative for the questions the authors are interested in asking. They try to control for this problem by comparing populations of a different species, but I was left unconvinced of the validity of this 'control'.

This also influences the 'Validity of Findings'.

Validity of the findings

This article does not meet the PeerJ standards for the 'validity of the findings'. Geography is confounded with the two forms of C. pipiens. Adding additional populations of each form would greatly improve the comparisons and control for geographic influences on genetic divergence.

Additional comments

This manuscript assesses the genetic divergence between feral and domesticated forms of the disease vector, Culex pipiens, using newly assembled transcriptome data for each form. These two forms differ dramatically in their behaviors and physiology even though they are genetically similar and morphologically indistinguishable. The general question and topic are very interesting, and the study system seems quite appropriate to addressing this important question.

To address this, the authors generated and compared de-novo transcriptomes from one population of each form. They then performed a series of analyses to identify genes and gene ontologies that show evidence of accelerated evolutionary rates between the two forms. To then tease apart differences simply due to geographic isolation from differences underlying the behavioral and physiological differences, the authors also included a comparison of two strains of another mosquito species, Culex quiquefaciatus, to provide a control or divergence baseline.

I was left unconvinced of this strategy, which really underlies the entire logic to the paper. An intraspecific comparison between two allopatric populations of the same form would be a much better approach to controlling for differences across the transcriptome in C. pipiens. Moreover, repeated comparisons among many populations from each form, where the same genes or gene ontologies are found, would be much stronger evidence that the comparisons provided here.

---

## Round 0.2 · accepted · Accept

Thank you for carefully considering the concerns raised by the reviewers on the previous version. The fact that only one population per mosquito form was used in the study is now clearly acknowledged and the associated conclusions formulated accordingly.